# Psychometric Properties of the Albanian Version of the Nursing Self-Efficacy Scale

**DOI:** 10.3390/healthcare10112232

**Published:** 2022-11-08

**Authors:** Blerina Duka, Alessandro Stievano, Rosario Caruso, Emanuela Prendi, Florian Spada, Gennaro Rocco, Ippolito Notarnicola

**Affiliations:** 1Department of Biomedicine and Prevention, University of Rome Tor Vergata, 00133 Rome, Italy; 2Department of Biomedical Sciences, Faculty of Medicine, Catholic University “Our Lady of Good Counsel”, 1000 Tirana, Albania; 3Centre of Excellence for Nursing Scholarship, OPI, 00173 Rome, Italy; 4Department of Clinical and Experimental Medicine, University of Messina, 98100 Messina, Italy; 5Department of Biomedical Sciences for Health, University of Milan, 20133 Milan, Italy; 6Health Professions Research and Development Unit, IRCCS Policlinico San Donato, 20097 San Donato Milanese, Italy

**Keywords:** confirmatory factor analysis, exploratory factor analysis, instruments, nursing competencies, self-efficacy

## Abstract

There are few tools to ascertain self-efficacy, which is a valuable component of nursing skills. This study has tested the psychometric properties of an Albanian translation of the Nursing Profession Self-Efficacy Scale (NPSES), which is based on Bandura’s theory of social cognition. Data were collected using questionnaires which were filled out by 423 nurses from the twelve provinces of the Albanian health system. The scale’s content, face, and construct validity were evaluated. Reliability was verified using Cronbach’s α and test–retest and by calculating the intraclass correlation coefficients. The original NPSES has two factors; for the A-NPSES analyzed in this study, four factors emerged from the factorial analysis of our reference sample: nursing care procedure situation, nursing research situation, nursing ethics situation, and nursing practice situation. Cronbach’s α was 0.91, indicating that the tool is reliable. The results of this study demonstrate the validity and reliability of the Albanian version of the NPSES. This scale is a valuable tool for self-assessing nurses’ self-efficacy. An adequate scale for measuring nurses’ self-efficacy can be used to improve the quality of clinical nursing care.

## 1. Introduction

To meet growing global health needs, the nursing profession has undergone several changes in recent years [1]. During these changes, various efforts have been made to better understand the modern nursing culture and nursing activities, particularly nursing performance [2].

Self-Efficacy (SE) derives from Albert Bandura’s [3] socio-cognitive theory and is defined as a person’s perception of their own ability to successfully perform a given task. According to Bandura [4], SE, by mediating the relationship between knowledge and behavior, could impact individuals’ actions and thoughts. For example, people with low SE may experience negative thoughts about their ability to achieve their goals due to low self-esteem and a sense that they lack competence in handling challenging tasks in different contexts [5]. Conversely, people with high SE are more likely to persevere in their efforts to achieve success and meet their goals [6].

Consequently, SE influences nurses’ actions, behaviors, performance, and the quality of their decisions [7,8]. Previous studies have shown that professionals’ SE plays an important role in improving performance and in improving professional clinical practice [9,10].

Self-efficacy (SE) has been identified as a factor that can influence nursing activities and nursing performance [11,12]. SE is therefore a fundamental aspect of the nursing profession, as it is closely linked to decisions taken in clinical nursing care [13]. SE is relevant to nursing activities because it reflects nurses’ ability to verify nursing practices in different clinical contexts and influences nurses’ performance [14,15,16].

SE offers a perspective on nursing care that, by combining practice and theory, identifies changes in nurses and in their responses to patients’ healthcare needs [17]. In addition, SE affects the relationship between knowledge and behavior and impacts nurses’ progressive development of skills, which can improve their clinical practices in different clinical contexts [18,19,20].

Therefore, it is necessary to measure nurses’ SE specifically; this approach involves identifying particular characteristics that impact nurses’ SE, which may include ethical values associated with patients’ well-being and the behaviors and norms that guide nurses’ practices and daily care in different clinical contexts [21,22].

Therefore, to measure nurses’ SE, a scale that reflects the specific characteristics of the nursing profession is needed to ensure accurate results [23]. The Nursing Profession Self-Efficacy Scale (NPSES) was originally developed in Italy by Caruso et al. [24] and this scale is based on Bandura’s theory of social cognition. The NPSES measures nurses’ professional activity, which includes evidence-based scientific knowledge and skills, ethical values, and relationships, as well as cooperation with peers to meet patients’ needs [24].

The purpose of the present study is to verify the validity and reliability of the Albanian version of the NPSES (A-NPSES), a translation and cultural adaptation of the original NPSES scale designed for the Albanian context. Validation of the NPSES scale in the Albanian context could have a major impact on the nursing management of both educators and health policy, as it provides a tool for exploring nurses’ confidence in addressing their job challenges. Furthermore, self-efficacy can also be used to predict nurses’ clinical activities and competence development.

## 2. Materials and Methods

### 2.1. Study Design

To verify the validity and reliability of the A-NPSES, the cultural adaptation and psychometric properties of the tool were evaluated.

### 2.2. Instrument

The NPSES, a scale that measures nurses’ professional SE, consists of 19 questions that measure the characteristics of care (12 items) and professional situations (7 items). Respondents can indicate their degree of agreement with each element of the NPSES scale using a 5-point Likert scale ranging from 1 (completely disagree) to 5 (completely agree). The total score for the scale is calculated from the sum of the individual item scores; the final score can range from 19 to 95, with higher scores indicating greater SE. Respondents’ demographic data are also collected, including gender, age, marital status, education level, educational background, department, work experience (years), experience in the department (years), and job satisfaction. The internal consistency of the original NPSES scale has a Cronbach’s α of 0.830.

### 2.3. Setting and Sample

The sample of our study consisted of nurses working in hospitals in one of the twelve Albanian provinces (Berat, Dibër, Durrës, Elbasan, Fier, Gjirokastër, Korcë, Kukës, Lezhë, Shkodër, Tiranë, Vlorë). Participants were recruited from May 2022 to July 2022 using convenience sampling. The study inclusion criteria were as follows: All participants were (a) nurses (b) working in a clinical setting who (c) provided informed consent to participate in the study. Registered nurses who worked in private clinics and those who worked in non-clinical (e.g., administrative) nursing settings were excluded from the study, as the inclusion of these participants could have an impact on outcomes. The optimal sample size for factor analysis was determined using the maximum likelihood method.

Previous studies have found that the optimal sample size should be the basis for the n:p ratios; in other words, the ratio between the number of cases and the measured variable can be 20:1 or 10:1 [25,26]. Since the NPSES scale includes 19 items, a sample of 190–380 participants was determined to be the optimal size. Therefore, 455 nurses who met the inclusion criteria were recruited. Of these, 32 were subsequently excluded due to incomplete questionnaires, leaving a final sample of 423 participants.

### 2.4. Ethical Statement

Ethical approval for this study was provided by the Ethics Committee of the Order of Albanian Nurses (1.12.2022).

The purpose of this study was explained to the nursing coordinators of the wards at each hospital where participants were recruited. The hospitals also agreed to participate in the study after approving our application to participate in the study. Before data collection, participants were informed about the study’s purpose, procedures, and research method, data anonymity, and the possibility of participation or withdrawal. Study contributors were then asked to read and sign the informed consent form.

### 2.5. Translation Process and Content Validity

The NPSE was translated into Albanian for the present study with the permission of the original developers of the scale [24]. The scale was translated from English to Albanian following the guidelines for forward–backward translation proposed by Beaton et al. [27]. The translation was performed by a bilingual (English and Albanian) professor of nursing science. The translation was then revised by an international interpreter and a nursing professional, both of whom have lived in English-speaking countries for over ten years. They later provided us with feedback on the expressions and clarity of the Albanian translation and assessed the adequacy of the translation. Subsequently, it was referred to 20 Albanian nurses, all with over five years of clinical nursing experience. The nurses confirmed their understanding of the elements of the scale and identified points that needed to be changed to address cultural differences. Subsequently, the reverse translation was performed by an English-speaking interpreter, and the similarities between the original text and the reverse translation were assessed by two native English speakers.

The content validity of the A-NPSES was confirmed by ten experts (four nursing coordinators and six nursing professors) using the Content Validity Index (CVI) to verify the tool’s relevance and clarity. The experts evaluated the Content Validity Ratio (CVR), the Content Validity Index for Scale (S-CVI), and the content validity index at the element level (I-CVI) [28]. First, the experts were asked to indicate whether each item in the scale was needed to make a constructed work in a set of elements of the scale. Each item was assigned a score of 1, ‘not necessary’; 2, ‘useful but not necessary’ or 3, ‘necessary’. The CVR for the content ranged from 1 to −1. A higher score indicated more agreement by the experts on the need to include the item in the scale. We defined the CVR using the formula CVR = (Ne − N/2)/(N/2), where N is the total number of experts. The numerical value of the CVR is indicated by the Lawshe table [29].

The CVR for the translated scale was greater than 0.62, and the items reached a significant level of acceptance [29]. The S-CVI and the I-CVI [28] ranged from −1 to +1. A value of 0.70 or higher is considered sufficient grounds to retain items in the translated version, as it was in our case [30].

### 2.6. Data Analysis

The data collected in our study were analyzed using the statistical software SPSS 24 (SPSS Inc., Chicago, IL, USA) and the statistical package R.

The sample (N = 423) was randomly divided into two groups. For group 1 (N = 211), an Exploratory Factor Analysis (EFA) was conducted; for group 2 (N = 212), Confirmatory Factor Analysis (CFA) was used. We analyzed differences in demographic characteristics and found no significant differences between the two groups.

An EFA was used to verify the validity of the construct. The Kaiser–Meyer–Olkin test and Bartlett’s sphericity test were used to verify that the processed data were appropriate for factor analysis; principal component analysis and varimax rotation were utilized to extract the factors [31].

The model fit indices were calculated using CFA. In general, a χ^2^/df ratio between 2 and 5 is considered acceptable even when the sample size is small; however, the Comparative Fit Index (CFI) is less affected by sample size than the χ^2^/df ratio. A fit index of at least 0.70 is considered acceptable, although an index of 0.90 or higher is preferred [32]. A Goodness-of-Fit Index (GFI) of 0.90 or higher indicates a good fit for the model [32,33].

However, as the model becomes more complex, the likelihood that the GFI will be affected by the sample size increases [33,34]. The Tucker–Lewis Index (TLI) is not affected by sample size, but its value must be considered in the assessment of suitability; a value over 0.90 is considered appropriate [35]. The Root Mean Square Residual (RMSR) is the mean value of all standardized residuals; it is used to test the proximity of fit in ordinal factor analysis. A well-fitting model has an RMR value of less than 0.05 [36].

The Root Mean Square Error of Approximation (RMSEA) is a commonly used index. Because this index is very sensitive to changes in sample size, additional goodness-of-fit measures have been included; here, values below 0.08 are considered acceptable [37].

We used Cronbach’s α to measure the internal consistency of each item and to verify the validity of each dimension of the A-NPSES scale. Typically, a Cronbach’s α coefficient over 0.90 indicates a high degree of confidence; a value over 0.80 indicates a normal degree of confidence, a value over 0.70 indicates an acceptable degree of confidence, and a value over 0.60 indicates a low degree of confidence. Therefore, reliability is typically confirmed if Cronbach’s α is greater than 0.70 for new instruments or greater than 0.80 for consolidated instruments [30].

## 3. Results

Most participants in our study were female (85.3%) and 20 to 41 years old (69.5%). The most common departments in which participants worked included medicine (65.2%), critical care (11.6%), and pediatrics (6.4%). Most (62.9%) of the sample had been working as nurses for zero to 11 years. Most worked in public health facilities (96.9%). Almost all of the participants (93.2%) indicated that they were quite satisfied or more satisfied with their work. Table 1 shows the demographic characteristics of the participants.

### 3.1. Face and Content Validity

In total, 20 panel members who were educational experts participated in this phase; of these, 70% (N = 14) were women. The panelists’ mean age was 49.70 years; their age ranged from 34 to 65 years.

Table 2 shows the validity indices for the experts’ evaluations of the A-NPSES. The I-CVI and S-CVI indicated satisfactory validity. The validity indices for the expert evaluations ranged from 0.80 to 1 for the I-CVI and from 0.80 to 0.87 for the S-CVI.

### 3.2. Construct Validity

We used an EFA to psychometrically evaluate the A-NPSES. The Kaiser–Meyer–Olkin test (0.909) and Bartlett’s sphericity test (χ^2^ = 1799.856, df = 153, *p* = 0.000) indicated that the data were appropriate for factor analysis [38]. The 18 items of the A-NPSES were divided into four dimensions and/or factors: Items 1 to 6 measured factor 1, items 7 to 9 measured factor 2, items 10 to 14 measured factor 3, and items 15 to 18 measured factor 4. Item 13 was excluded from our version as it had low communalities (0.376) [39]. These four factors explained 62.96% of the overall variance. Although some items had cross-loads with similar loads between the factors, overall, this solution demonstrated a good structure [40]. The factor loadings are shown in Table 3.

The confirmatory model produced the following values: χ² = 342.41; 129 degrees of freedom (df); χ²/df = 2.65 and *p* = 0.000. The CFI was 0.91 and the TLI was 0.89. The RMSEA value of 0.09 (IC = 0.8 ± 1.0) showed a discrete fit for the four-factor model, as reported in Figure 1.

The model explained 62.96% of the total variance. Factor 1 accounted for 41.75% of variance, factor 2 for 8.31%, factor 3 for 7.22%, and factor 4 for 5.67%. The A-NPSES also showed adequate internal consistency for each domain and for the overall scale. For factor 1, α = 0.763; for factor 2, α = 0.810; for factor 3, α = 0.772; for factor 4, α = 0.788. Cronbach’s α for the overall scale was 0.911 (Table 4).

Based on the resulting pattern matrix of the EFA, a four-factor model could be described. Items 4, 19, 3, 15, 5, and 11 were included in factor 1 and labeled nursing care procedures, while items 10, 12, and 11 were included in factor 2 and labeled nursing research situation. Items 8, 7, 9, 2, and 6 were included in factor 3 and labeled nursing ethics situation. Finally, items 17, 16, 14, and 18 were included in factor 4 and labeled nursing practice situations (Table 3).

## 4. Discussion

This study described the psychometric validation of the A-NPSES for measuring the SE of registered nurses working in different clinical settings in Albania. The A-NPSES tool includes 18 elements that are evaluated using Likert scales. The A-NPSES was assessed before the cross-sectional data needed to estimate the validity of the construct. Expert evaluations of the 18 items of A-NPSES indicate that the items are adequate. To improve nursing care, it is critical to comprehend the impacts of SE on practice, professional behavior, identity, and professional culture [41,42,43]. The A-NPSES can be considered an adequate tool for assessing SE in different clinical, educational, and research contexts in Albania.

SE, which is the theoretical foundation for the original NPSES scale developed by Caruso et al. in 2016, can directly influence nursing skills and, consequently, the behavior of nurses in clinical and professional environments around the world [14,44]. The original NPSES developed by Caruso et al. (2016) [24] includes two factors: ‘quality of service’ and nursing ‘professional situations’; these factors measure the skills nurses need to work in professional ways. Our study is more similar to a previous study located in Korea [45] and defined four factors in the A-NPSES.

The four factors in our scale can be described as follows: Factor 1, the nursing care procedure situation, mainly addresses SE related to nurses’ skills in various clinical situations. Factor 2, the nursing research situation, mainly addresses SE associated with research-based nursing practices. Factor 3, the nursing ethics situation, addresses SE related to nursing ethical issues. Factor 4, the nursing practice situation, addresses SE related to different nursing practices in different clinical contexts. These factors were defined based on the nursing activities described in the A-NPSES after careful discussion by the authors.

We developed a fit model using CFA that showed that a four-factor structure explained the data collected from our reference sample better than the original two-factor model. Surely, the differences in the characteristics of the sample analyzed and the different cultural contexts are attributable to different nursing environments and may explain the differences in the structure of the factors, as they also indicated in the study of the Korean researchers, where even the structure of NPSES in the Korean context was to four factors [45].

In addition, the factoriality of each dimension of the A-NPSES was supported by good internal consistency, indicated by Cronbach’s α value of 0.911. In contrast, for the original NPSES, Cronbach’s α is only 0.830 for the overall scale, while Cronbach’s α for each item ranges from 0.904 to 0.913, and Cronbach’s α for each factor ranges from 0.763 to 0.810.

The factor that explained the highest percentage of variance in our scale was the nursing research situation, followed by the nursing practice situation, nursing ethics situation, and nursing care procedure situation. For our four-factor model, the fit indices of CFI (0.909), TLI (0.892), and GFI (0.845) indicate a very good fit; the RMSR (0.031) meets the recommended index, and the RMSEA (0.089) indicates approximately 90% confidence. The population RMSEA for the default model is between 0.077 and 0.100. Therefore, the results of the adaptation index indicate good adaptation for all four factors.

The A-NPSES could help improve awareness of the role of SE in current nursing practice and help nurses better understand their goals within national health systems. To effectively evaluate nursing practices, it is essential to examine how SE impacts nurses’ professional identity and adherence to standards of competence in clinical practice. It is also important to examine the relationship of SE with nursing skills and to explore how SE can change nurses’ behaviors in clinical practice [46].

However, more research is needed to verify the four-factor structure of the A-NPSES with a larger sample of Albanian registered nurses.

### Limitations

One of the main limitations of this study is that participating nurses from the twelve Albanian provinces were selected using convenience sampling. This could negatively impact the representativeness of the sample and, therefore, the generalizability of the results and may have created sampling distortions that could negatively impact the external validity of the results.

A limitation of our study may have been the choice of the ten experts, as they seem to be a small sample to evaluate CVI; a larger number of participants involved in the evaluation of CVI could have given more significance to CVI.

Another limitation could be the consideration of cross-loads; future research should investigate this aspect more in future psychometric assessments of this scale.

In addition, future research should examine the performance of the A-NPSES over time. In this study, we assessed the reliability of the scale by evaluating the internal consistency of the factors, but no data on the scale’s stability over time were collected.

## 5. Conclusions

The results of this study confirm that the validity and reliability of the A-NPSES are acceptable. Cronbach’s α for the scale’s content, criterion, and construct validity indicates that the scale is useful for measuring the SE of Albanian nurses. Therefore, the A-NPSES may be considered a useful tool for measuring SE; moreover, it could also be useful for research investigating the relationship of SE with nursing behaviors and outcomes.

The present results also suggest that the NPSES may have different factorial structures in different cultures, a finding that aligns with previous studies [41]; however, further investigations should be conducted to establish the validity of this tool in other social contexts.

## Figures and Tables

**Figure 1 healthcare-10-02232-f001:**
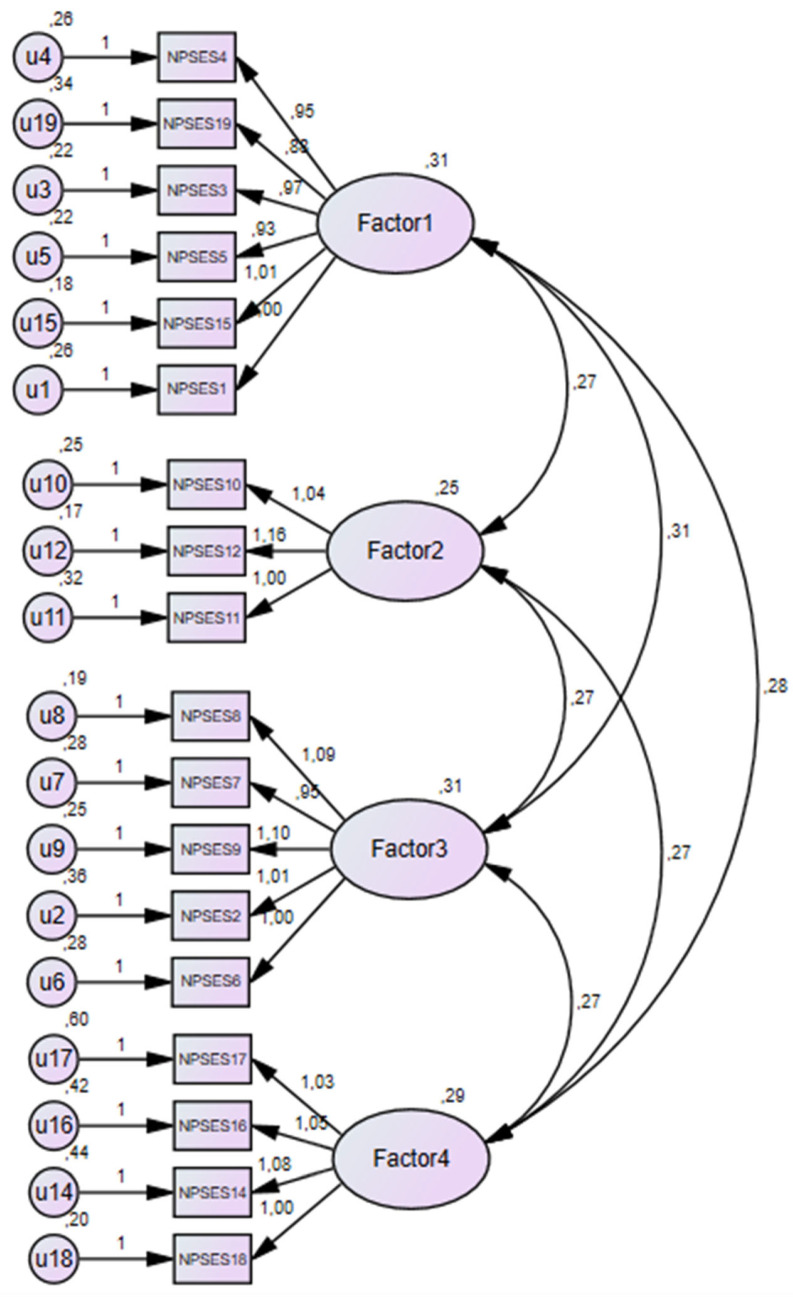
Structure of the Albanian version of the NPSES based on confirmatory factor analysis.

**Table 1 healthcare-10-02232-t001:** Demographics statistic (N = 423).

	N	%
Gender		
F	361	85.3
M	62	14.7
Age Classes		
20–30	161	38.1
31–41	133	31.4
41–52	82	19.4
53–63	47	11.1
Work Place		
Private	13	3.1
Public	410	96.9
Work Unit		
Critical Area	49	11.6
Medical Area	276	65.2
Pediatric Area	27	6.4
Undeclared	71	16.8
Years Profession		
0–11	266	62.9
12–23	102	24.1
24–35	48	11.3
36–47	7	1.7
Years Ward Unit		
0–9	288	68.1
10–19	80	18.9
20–29	40	9.5
30–39	15	3.5
Job Satisfaction		
Nothing	8	1.9
Little	21	5.0
Fair	180	42.6
More	214	50.6

**Table 2 healthcare-10-02232-t002:** Content validity of A-NPSES (I-CVIs and S-CVI) (N = 10).

	I-CVIs	Interpretation	S-CVI
NPSES 1	0.82	Pertinent	0.87
NPSES 2	0.82	Pertinent	
NPSES 3	0.80	Pertinent	
NPSES 4	0.96	Pertinent	
NPSES 5	0.85	Pertinent	
NPSES 6	0.80	Pertinent	
NPSES 7	0.94	Pertinent	
NPSES 8	0.86	Pertinent	
NPSES 9	0.90	Pertinent	
NPSES 10	0.80	Pertinent	
NPSES 11	0.83	Pertinent	
NPSES 12	1	Pertinent	
NPSES 13	0.80	Pertinent	
NPSES 14	1	Pertinent	
NPSES 15	0.94	Pertinent	
NPSES 16	0.84	Pertinent	
NPSES 17	0.88	Pertinent	
NPSES 18	0.85	Pertinent	
NPSES 19	0.85	Pertinent	

**Table 3 healthcare-10-02232-t003:** A-NPSES exploratory factor analysis ^1^.

	New Item Scale A-NPSES	Old Item Scale NPSES	F1	F2	F3	F4	Communalities
Factor 1 Nursing Care Procedure Situation	NPSES1	NPSES4	**0.713**	0.164	0.371	0.130	0.628
	NPSES2	NPSES19	**0.649**	0.009	−0.075	0.384	0.629
	NPSES3	NPSES3	**0.642**	0.374	0.321	0.068	0.660
	NPSES4	NPSES5	**0.629**	0.223	0.450	0.033	0.689
	NPSES5	NPSES15	**0.607**	0.605	0.030	0.114	0.649
	NPSES6	NPSES1	**0.606**	0.078	0.456	0.216	0.510
Factor 2 Nursing Research Situation	NPSES7	NPSES10	0.108	**0.786**	0.162	0.150	0.503
	NPSES8	NPSES12	0.307	**0.755**	0.235	0.000	0.679
	NPSES9	NPSES11	0.090	**0.696**	0.282	0.397	0.558
Factor 3 Nursing Ethics Situation	NPSES10	NPSES8	0.115	0.273	**0.760**	0.120	0.679
	NPSES11	NPSES7	0.240	0.153	**0.607**	0.229	0.730
	NPSES12	NPSES9	0.134	0.464	**0.570**	−0.023	0.720
	NPSES13	NPSES2	0.321	−0.009	**0.566**	0.453	0.572
	NPSES14	NPSES6	0.271	0.238	**0.481**	0.385	0.748
Factor 4 Nursing Practice Situation	NPSES15	NPSES17	0.105	−0.014	0.118	**0.774**	0.704
	NPSES16	NPSES16	0.281	0.292	0.161	**0.716**	0.623
	NPSES17	NPSES14	0.010	0.476	0.188	**0.556**	0.479
	NPSES18	NPSES18	0.453	0.174	0.180	**0.459**	0.575

**^1^** Extraction Method: Principal Component Analysis. Rotation Method: Varimax with Kaiser Normalization. Rotation converged in 9 iterations.

**Table 4 healthcare-10-02232-t004:** Cronbach’s alpha of NPSES.

		Mean	SD	Corrected Item-Total Correlation	Cronbach’s Alpha If Item Deleted	Cronbach’s Alpha Factor	Total Cronbach’s Alpha
Factor 1 Nursing Care Procedure Situation	NPSES1	4.4929	0.64269	0.652	0.905	0.763	0.911
	NPSES2	4.4408	0.74963	0.431	0.911		
	NPSES3	4.5261	0.61955	0.670	0.905		
	NPSES4	4.5877	0.59812	0.631	0.906		
	NPSES5	4.6066	0.59502	0.639	0.905		
	NPSES6	4.5403	0.67756	0.636	0.905		
Factor 2 Nursing Research Situation	NPSES7	4.4028	0.73276	0.540	0.908	0.810	
	NPSES8	4.5024	0.69264	0.598	0.906		
	NPSES9	4.2796	0.77614	0.673	0.904		
Factor 3 Nursing Ethics Situation	NPSES10	4.5308	0.64936	0.584	0.906	0.772	
	NPSES11	4.3318	0.71307	0.555	0.907		
	NPSES12	4.4929	0.69261	0.525	0.908		
	NPSES13	4.4123	0.75318	0.607	0.906		
	NPSES14	4.3033	0.72568	0.621	0.905		
Factor 4 Nursing Practice Situation	NPSES15	4.1469	0.93209	0.403	0.913	0.788	
	NPSES16	4.3412	0.83218	0.657	0.904		
	NPSES17	4.2322	0.92998	0.531	0.909		
	NPSES18	4.5355	0.62672	0.577	0.907		

## Data Availability

The data presented in this study are available within the article.

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
