# Peer review of "Psychometric Properties of the Albanian Version of the Nursing Self-Efficacy Scale"

_healthcare, 2022, doi:10.3390/healthcare10112232_

Round 1

Reviewer 1 Report

First of all, I want to congratulate the authors for the study. In fact, the article reports an important study for the academic and scientific community. The study presents scientific and methodological criteria. After some changes, which are described later, it must be accepted for publication.  

Please answer and apply in the document the next questions:

1. Lines 68-70 – Does this information make sense in the introduction? 2. Regarding to the ethical and deontological principles of scientific research, why the ethics committee requests did not requested in the hospitals (the real context)?

3. Lines 129 and 137 – (Polit & Beck, 2006)- References: plese see the information – apply in the entire document:

“In the text, reference numbers should be placed in square brackets [ ], and placed before the punctuation; for example [1], [1–3] or [1,3]. For embedded citations in the text with pagination, use both parentheses and brackets to indicate the reference number and page numbers; for example [5] (p. 10). or [6] (pp. 101–105).

4. Lines 147-150 – references used? 5. Lines 173-178 - these characteristics may condition the results of the study, and should be addressed in the limitations of the study. What is your opinion? 6. Lines 222-224 - Are there no more references that can support this consideration?

7. In the topic named limitations, are there no further limitations of the study?

Author Response

R1 comments:

First of all, I want to congratulate the authors for the study. In fact, the article reports an important study for the academic and scientific community. The study presents scientific and methodological criteria. After some changes, which are described later, it must be accepted for publication. Please answer and apply in the document the next questions:

Thanks for your valuable and essential comments. They were very useful to improve the paper!!

  1. Lines 68-70 – Does this information make sense in the introduction?

Based on the reviewer's suggestion, we removed the sentence as the information was not essential and did not provide further evidence.

  1. Regarding to the ethical and deontological principles of scientific research, why the ethics committee requests did not requested in the hospitals (the real context)?

Based on the reviewer's suggestion, we have included a sentence that improves the text for the journal's readers

  1. Lines 129 and 137 – (Polit & Beck, 2006)- References: please see the information – apply in the entire document: “In the text, reference numbers should be placed in square brackets [ ], and placed before the punctuation; for example [1], [1–3] or [1,3]. For embedded citations in the text with pagination, use both parentheses and brackets to indicate the reference number and page numbers; for example [5] (p. 10). or [6] (pp. 101–105).

Based on the reviewer's suggestion, we have inserted the bibliographic references within the text, and we have also arranged the subsequent numbering of the bibliographic references.

  1. Lines 147-150 – references used?

Based on the reviewer's suggestion, we have included the bibliographic references

  1. Lines 173-178 - these characteristics may condition the results of the study, and should be addressed in the limitations of the study. What is your opinion?

We thank the reviewer for this idea. In our opinion, these characteristics complement the study by describing the results in the best possible way by providing the reader with a complete picture of the sample we have analyzed; for this reason, we have chosen to include them  at the beginning of the Results paragraph. Last, we do not believe that they can be a source of limitation for our study.

  1. Lines 222-224 - Are there no more references that can support this consideration?

As recommended by the reviewer we have added some bibliographical references to supplement the concept expressed in the sentence.

  1. In the topic named limitations, are there no further limitations of the study? Less...

Based on the reviewer's suggestion, we included other limitations of our study

Reviewer 2 Report

Dear authors, thank you for allowing me to revise this manuscript describing the validation process of the Nursing Self-Efficacy Scale. Although I found the manuscript was well written and the study extensively conducted in all the required psychometric tests, there are some points of concern that I would discuss with you.

INTRODUCTION - well written and sustained by relevant citations. I missed the rationale to validate the instrument in Albanian; why did you perform this study?

Minor - page 2, line 61. Please, replace "job" with "practice/profession".

METHODS - I missed some details about the Albanian context. On page 2, lines 92-94, you stated "registered": does Albania has a National registry of nurses? It's unclear what you meant with "on the ground"; finally, why may your results be impacted by the presence of such nurses in your sample?

Page 3, line 125.  Ten experts seems a poor sample to assess CVI. The latest guidelines released by the COSMIN consortium recommended a higher number of participants involved in the Content validity assessment. You should insert this in your limitations section.

RESULTS - In the first line, the percentage (54.6%) is not consistent with those reported in Table 1. Moreover, in such a table, there is a heading labelled "ins privato"; I presume it is a typo.

In the heading of table 2, please provide your experts' sample (n=10).

Page 5, line 191. Why items are reduced to 18? I am sure you haven't said that you removed an item (the 13th of the old scale) after the content validity assessment. 

Table 3 provided factor loadings for each item. And some of them have shown cross-loadings; in some cases, values are very close, as for items 15, 9, 2, 14, and 18 of the old scale. I would expect a clarification on this issue in the discussion and recognition as a potential limitation in the appropriate section. 

DISCUSSION - I found this section as a repetition of the results. It seems to me that it needs to discuss why the scale you have validated differs so much psychometrically from the Italian version; could there be cultural differences in the sample that led to these different psychometric properties?

Are there differences in the nursing curriculum or cultural aspects in your population that may lead us to think of a different characterization?

A discussion should be made concerning cross-loading in the EFA that may be a symptom of a less-than-perfect fit of the scale. Or, more appropriately, a hard-to-assess structure of this scale?

Minor - page 9, line 262. Please, replace "higher" with "larger".

I hope my suggestions will help you improve your study quality.

Author Response

R2 comments:

Dear authors, thank you for allowing me to revise this manuscript describing the validation process of the Nursing Self-Efficacy Scale. Although I found the manuscript was well written and the study extensively conducted in all the required psychometric tests, there are some points of concern that I would discuss with you.

First of all, thanks for your valuable and fundamental comments and also for the opportunity to improve the paper!!

INTRODUCTION - well written and sustained by relevant citations. I missed the rationale to validate the instrument in Albanian; why did you perform this study? Minor - page 2, line 61. Please, replace "job" with "practice/profession".

Based on the reviewer's suggestion, we added a rationale to better explain the usefulness of validating the NPSES in the Albanian context. In this way we hope to have improved the readability of the paper Also, as advised by the reviewer, we changed the word "job" with "profession".

METHODS - I missed some details about the Albanian context. On page 2, lines 92-94, you stated "registered": does Albania has a National registry of nurses? It's unclear what you meant with "on the ground"; finally, why may your results be impacted by the presence of such nurses in your sample? Page 3, line 125. Ten experts seems a poor sample to assess CVI. The latest guidelines released by the COSMIN consortium recommended a higher number of participants involved in the Content validity assessment. You should insert this in your limitations section.

Based on the reviewer's suggestion, we rewrote the sentence, trying to improve its understanding for the journal's readers. Yes, in Albania there is a national register of nurses and a national regulatory authority (UISH). We selected ten experts according to the recommendations of Lawshe et al. as quoted in the text, however based on the reviewer's suggestion, we have included a sentence within the limitation section to better explain the issue.

RESULTS - In the first line, the percentage (54.6%) is not consistent with those reported in Table 1. Moreover, in such a table, there is a heading labelled "ins privato"; I presume it is a typo. In the heading of table 2, please provide your experts' sample (n=10). Page 5, line 191.

Why items are reduced to 18? I am sure you haven't said that you removed an item (the 13th of the old scale) after the content validity assessment. Table 3 provided factor loadings for each item. And some of them have shown cross-loadings; in some cases, values are very close, as for items 15, 9, 2, 14, and 18 of the old scale. I would expect a clarification on this issue in the discussion and recognition as a potential limitation in the appropriate section.

Based on the reviewer's suggestion we corrected the results. It was our transcription error. We have also included in the header the number of experts, as indicated by the auditor. We also added a sentence to specify as requested by the reviewer the exclusion of item 13. In fact, it had a low commonality, as indicated in the paper. Regarding the cross-loadings we have inserted a sentence to improve the understanding of this concept. We also included in the limitations section what could be the future implications of this study.

DISCUSSION - I found this section as a repetition of the results. It seems to me that it needs to discuss why the scale you have validated differs so much psychometrically from the Italian version; could there be cultural differences in the sample that led to these different psychometric properties?

Are there differences in the nursing curriculum or cultural aspects in your population that may lead us to think of a different characterization? A discussion should be made concerning cross-loading in the EFA that may be a symptom of a less-than-perfect fit of the scale. Or, more appropriately, a hard-to-assess structure of this scale? Minor - page 9, line 262. Please, replace "higher" with "larger". I hope my suggestions will help you improve your study quality.

Based on the reviewer's suggestion, we have included a sentence that improves understanding of the four-factor structure of our scale also in respect with cultural differences. We have also included another point within the limitations concerning cross-loading in the EFA. Finally, as recommended by the reviewer, we also changed the word "higher" to "larger",

Round 2

Reviewer 2 Report

Dear authors, thank you for having considered my comments. I can see these improvements to your manuscript and am satisfied with this second version.